# "Since I've Been Ill, I Live Better": The Emergence of Latent Spirituality in the Biographical Pathways of Illness

Nicola Luciano Pannofino

Department of Cultures, Politics and Society, University of Turin, Turin 10153, Italy; nicolaluciano.pannofino@unito.it

**Abstract:** Spirituality can be a crucial resource to draw on to make sense of critical situations that mark a turning point in individual and collective biographies. In these cases, a ritual and symbolic response to the trauma may occur, bringing to the surface a «latent spirituality», that is, a tacit propensity towards the sacred that manifests itself in unexpected ways, even in those who do not normally believe or practice, in extraordinary situations that engender fear, anomie or disorientation and that have profound existential repercussions. This article aims to investigate the latent spirituality in the face of the critical event represented by the onset of severe disease, based on the analysis of spiritual illness narratives collected in Italy through qualitative interviews with oncological patients. The narratives show how the condition of suffering can bring to light an unexpressed spirituality, consisting of the revitalization of previous traditional faith or the elaboration of an innovative lay spirituality. Data confirm how pathology constitutes a biographical fracture, accompanied by questions and needs of a religious and spiritual nature. In contrast to the prevailing approach in the medical humanities where spirituality is interpreted as a starting resource to which sufferers resort to cope toward the disease, these results indicate that the spiritual dimension is an emerging aspect along the therapeutic pathway and that it is transformed by reflecting the temporality of the biographical experience of illness.

**Keywords:** biographical fracture; illness narrative; latency; oncology; spirituality; time

## 1. Introduction

Spirituality can be a crucial resource to draw on to make sense of critical situations that mark a turning point in individual and collective biographies (Harper and Pargament 2015). This is what happens, for example, after the occurrence of environmental disasters in which spiritual support can help develop resilience capacities on the part of affected communities (Cherry et al. 2018). Indeed, recourse to spirituality can open up perspectives of meaning by which disaster survivors overcome the suffering of the present moment and re-establish a state of well-being by leveraging feelings of hope and group cohesion (Akbar 2019; Lalani et al. 2021). A focus on the spiritual dimension in crisis experiences enables caregivers to design interventions with the aim of supporting affected populations and coping in the short- and long-term post-traumatic phases (Captari et al. 2023), according to guidelines prepared for physicians, nurses, psychologists, religious groups and humanitarian and voluntary associations such as the Red Cross working in emergency settings (Roberts and Ashley 2008).

These studies agree that spirituality is a resource available for individuals to draw on in developing coping strategies in the face of traumatic events. An aspect less explored in the literature is how such events affect spirituality and what transformative impact they have on it. Thus, in the face of an extraordinary and unforeseen event, a ritual and symbolic reaction to the trauma may occur that brings to the surface a latent spiritual capacity (Price 1972), that is, a tacit propensity toward the sacred that manifests itself, even in individuals who do not habitually believe or practice; this propensity is expressed in critical occasions

that provoke fear, anomie, or disorientation and that have a relevant existential impact. The latent dimension of spirituality, in this sense, has so far received little attention in sociology and the medical humanities.

The recent COVID-19 pandemic has made it possible to highlight, in a way that is unprecedented globally, the close link between crisis situations and spiritual needs. Studies conducted in different national and religious contexts have underscored the decisive supportive role played by spirituality in providing responses to the insecurities caused by the contagion during the health emergency. While the pandemic exacerbated states of existential distress (Upenieks 2022) and vaccine skepticism motivated by religious reasons (Zarzeczna et al. 2023), spirituality proved useful in counteracting emotions of fear, anxiety, loss of hope (Buchtova et al. 2022), depressive symptoms (de Souza et al. 2023) and stress (Kosic et al. 2023), in grieving the loss of a family member (Biancalani et al. 2022) or in fostering individual practices such as praying at home (Apergis et al. 2023), with consequences differentiated according to a wide variety of factors such as age (Keisari et al. 2022), socioeconomic status (Safdar et al. 2023) or health status (Durmuş and Durar 2022). But survivors of such crisis situations and dramatic episodes may activate sensemaking processes that involve a revision of one's value orientation, stimulating innovative spiritual reflection on life and death, the effect of which is a change, even a profound one, in personal priorities and aspirations (Lee et al. 2022). Some research on war veterans, for example, indicates that battlefield experience may be a trigger factor that prompts the acquisition of a previously unexpressed spirituality in military personnel, giving scientific grounding to the popular adage "there are no atheists in foxholes" (Snape 2015). Although this finding is controversial and, especially in secularized cultural contexts, even traumatic experiences may not result in spiritual feelings, beliefs or practices (Granqvist and Moström 2014), fear of death is nonetheless one of the main drivers for the emergence of such a latent spiritual attitude, regardless of the high or low level of self-reported religiosity (Willer 2009). A study based on a survey administered to World War II veterans, 50 years after the conflict, confirms an increase in the practice of prayer and more intense participation in rituals by respondents, particularly in the sample of former soldiers who report more traumatic experiences and a more polemical stance toward war (Wansink and Wansink 2013).

This article aims to investigate the latent spirituality in the face of the critical event represented by the onset of severe pathology such as cancer from the analysis of illness narratives collected with the "time box"—a creative research technique specially developed for this study—in two groups of lung cancer and breast cancer patients affiliated with two oncology associations in Italy. The narratives are divided into two types, corresponding to the forms of latent spirituality that emerge in the pathways of cancer illness: the rediscovery of prior religiosity and the acquisition of a new secular spirituality. These results are discussed, and suggestions for the application of the concept of latent spirituality in the practice of spiritual care in the clinical setting are provided in the conclusion.

## 2. Results—The Illness Narratives of Cancer Patients

The following pages illustrate some of the illness narratives collected through qualitative interviews using the creative time box technique and conducted with patients belonging to the two cancer associations. Among the 20 narratives collected, the most exemplary ones have been selected here, as they express the main themes that recur in the corpus of spiritual narratives of the cancer patients who participated in the study. These illness narratives show how the critical and unexpected experience of the disease constitutes a turning point that profoundly affects the patients' biography, bringing to the surface a latent spirituality. This spirituality, while it may draw on the narrator's prior religious faith, is usually not reduced to it, being a specific and innovative outcome of the illness journey. The following stories recount two main forms by which latent spirituality finds expression: (a) the rediscovery of initial religiosity, with more or less critical personal reworking; (b) the acquisition of a new secular, non-denominational spirituality.

*2.1. The Rediscovery of Prior Religiosity*

The first form by which the latent spirituality of the cancer patients examined manifests itself is the rediscovery or revitalization of prior religiosity, experienced with detachment until the time of illness. This specific form is found in the narrative of P.B. (a woman with lung cancer), a Catholic-educated, believing but non-practicing patient. Her rediscovery of Catholicism starts from the consideration "illness takes away the future". These are words that denounce a perception of time as a scarce resource in the face of the possible inauspicious outcome of pathology (Fitzpatrick et al. 1980) and testify to the contraction of the time horizon within which illness narratives move (Giarelli and Venneri 2009). This is the starting point that triggers in the interviewee a reflection on biographical time:

I realized that for all these years in the office I had no particular hobbies, I didn't know what to do, I couldn't watch television, I couldn't read [...]. Sometimes I have regrets because one says "when I maybe retire or get old later", we used to say with my husband, we can go here and there, and I regretted not doing it when I could, because now I don't know if I will be able to do everything I had planned in my mind.

The awareness of past time as "lost" time counterbalances the fear of not having enough time in the future to fulfill her dreams and desires. To cope with this fear, P.B. resorted to religion several times: initially, she turned to Buddhism and relaxation techniques, as suggested to her by some of her colleagues, but without gaining the hoped-for benefits. She then realized that she must look within herself for the right motivation to relate to the illness, "because I needed to spend hours at home alone [...] and this allowed me to regain hope, to regain strength". In addition to introspection, the interviewee found support in her Catholic faith:

I went to elementary and middle school in a nunnery, so they kind of directed me in that direction. Then I didn't attend anymore [...]. At the end of the day I believe in it because it helps me to have support. In this period I realized that maybe a prayer or these roses that my friends brought me, a thought that maybe a person who brings me this bracelet that a friend of mine got me a little Madonna, here they gave me and are giving me support in this period [...]. This relationship with religion has always been there but maybe in this period I feel it more.

In particular, the rediscovery and strengthening of the Catholic faith in which she was born and raised is expressed in her devotion to Saint Rita of Cascia, the saint of "impossible causes":

As a child I went to school with the nuns however I did not know Saint Rita and there were a number of things that led me to this saint [...]. In May my colleagues came in turn and filled my house with these roses of St. Rita, which are blessed, and which I didn't even know what they were.

Cancer prompted P.B. to recover her faith, while assigning new meanings and values to it. This religious component is part of a more general and profound reevaluation of life time, which is the paradoxically ameliorative effect brought about by the disease itself:

[The illness] has given me the opportunity to live more quietly, with my family, live more outdoors, it has given me a better life in that sense, I have found benefit, freer, less stressful. I have learned to live more by the day, I enjoy the things of the day more [...]. During this period I gained in quality of life, because I used to be very stressed. I used to go out at 8 o'clock in the morning and arrive at 6:30 at night, so now I don't have these stress problems anymore. My desire was to go for walks but I couldn't do them because I didn't have time, and instead now every afternoon I go for walks, so I'm living in a different way that maybe I like even more.

This theme is reiterated by one of the handwritten notes contained in the time box, which features the phrase "the life I'm living now is stress-free", which preludes a medita-

tion on the value of time lived and a comparison with past time, heterodirected by work rhythms, since "I'm very framed, I'm an accountant, I've lived the years with deadlines". Cancer, therefore, provided a valuable learning opportunity for the interviewee.

In the story of C.G. (a woman with breast cancer), "spirituality immediately after surgery was quite present":

> Then spirituality took over, because I'm Catholic anyway but not a frequenter and kind of like everybody when you need you pray. You ask for help from people close to you and you also ask for help from your religion that tells you to turn to your saints and in those moments you believe again. "I also recovered thanks to my daughter", she told me, "I understood the seriousness that you were running, I went to the [church of] Consolata", she was an avid smoker, and she vowed, "if my mom does well I will stop smoking". She took the little picture and brought it to me, I have it inside the box. And so we also regained a little bit of spirituality by clinging where we could. And then my breasts were reconstructed in 2008 and in 2009 he offered me the trip to Medjugorje and I have a little bag of the things collected there. Every now and then I drop by the Consolata, maybe I don't go every Sunday, however I pass by a church, I go in and say a prayer.

The illness induced the interviewee to a rediscovery of the Catholic faith in which she grew up but from which she had distanced herself over the years, as "there are times that you leave the church alone and times that you get reacquainted, and for me it was this time":

> [Religion] helps me in times when I have problems or things that are not going the right way. I take and go to the Consolata, I go to this little chapel, underneath I feel good it seems to give me more strength to understand my mistakes [...], it's a moment of meditation that I can only do there. At home sometimes I say maybe I take a little picture and pray, then I say to myself, "I'm tired I don't feel like it I'll think about it ask" and instead if I pass in front of a church I go in, maybe just for a few minutes and it does me more good that. Not as frequently but I have these moments when I need to go, I feel better, I give myself more strength.

Illness does not always strengthen faith; sometimes, indeed, it undermines initial religious convictions, as in the case of T.S. (a woman with breast cancer), who included among the items in the time box an image of Jesus as a symbol of her religiosity:

> [Faith] I had it before, however after I got sick it went more and more. I always went on Sundays to church. A friend of mine had given me the handkerchief of Lourdes, and then I would put it on my breast when I was having treatments, before the surgery. Then I was given holy cards, little madonnas, and I always attached them to my bra. Faith helped me so much, in the sense that I used to cling to the little Madonna of Lourdes to get me through the therapies, visits and so on.

T.S.'s story exemplifies the form of religiosity that emerges in the therapeutic journey of cancer patients connoted by a personal reworking of prior faith, a critical position towards the Church and religious institution that leads to a "my way" Catholicism (Garelli 2020). In fact, the interviewee described the present as a time of doubts about her faith, not finding in it an adequate answer to the questions she asks herself about the why of the illness experience she is facing:

> I've kind of drifted away now, though, because sometimes I have discouragements, saying, how come it's always everything to me? And the illness, and the death of my father, and the separation. I say [to God], "can't you look away please?" So I kind of believe, and I talk to him and I don't talk to him, in the sense that I'm a little wary. I say: "with so many bad people always there, squeezing on people who have already suffered in life because you have to keep inflicting on me all the time". I talk to him. "I don't go to church anymore because I don't believe in church but with him I have a dialogue".

Rather, it is in the illness itself that T.S. finds stimulation to face the difficulties and obstacles of life: "[the illness] has given me so much strength, so much self-confidence".

A similar thought was expressed by V.S. (a woman with breast cancer), who before the disease regularly attended church every Sunday. After cancer, however, she had questions that undermined the solidity of her religious convictions:

The question is: why does this have to happen to me? What have I done wrong? Who gave me this gift? So I didn't want to go to church anymore, I felt abandoned. Then I went to our pastor and he said, "you can't think that everyone and everybody has to be well, otherwise we would all be good, we would all go to heaven and life would be flat". I have to say, though, that wasn't enough for me. Although I would like to go to Lourdes [...]. My daughter told me that this place gives a special feeling; it is an experience that would be worth doing. In my opinion they gave me a lot, I never saw myself helping a sick person and instead now I really feel projected and it makes me feel good. I felt a quality that I didn't have, that maybe was hidden and I didn't get a chance except in this circumstance to try it.

If the illness has weakened her Catholic faith, it nevertheless allowed her to discover a "hidden" quality that encourages her to approach others' suffering spiritually. But the experience of illness also opened the respondent to the possibility of reclaiming time in daily life, placing herself at the center of a renewed focus on personal well-being. The focus on self recalls the phrase engraved on the bracelet given to her by her daughters for her birthday and included in the time box: "from today I think of myself". The bracelet is an object that she no longer wore, as well as two other objects she would have liked to have included in the time box, namely the pajamas used during her hospitalization and the wig she wore after losing her hair to chemotherapy, as if these objects "jinxed" her because they were linked to a sad period of the disease. Recapturing the time of life charts, a new path infuses deeper motivations to live the present and future time:

Cancer is a closed-box thing. I had the surgery and I say it ends there, they take a piece out and it ends, then after that there's a whole series of things that I hadn't evaluated, maybe nobody knows how it progresses. So it really marks a path for you that is mandatory, you can't get out of there. Whereas now that I can get out I find it a very interesting thing that therefore also produced something more for me. I am definitely someone who likes to live a lot, but this thing has relaunched me even more, and especially I would like to see these years—I have to wait for my husband to finish working to do our entertainment—I would like them to be already past, even if they age me, I would like them to be already past to see how they are afterwards, because I would hate for you [the husband] to keep working and I get worse and we cannot finish our path.

L.V. (a woman with breast cancer) recounts the beginning of her illness journey as marked by disorientation, expressed by the question, "but why are you mad at me?"—a question from which illness stories often move (Charon 2006), turning to God, in an inner relationship that accompanied the illness experience and constituted a way of making sense of what happened to her. As the interviewee states:

I was talking face to face with God. I didn't go to church, I didn't talk to priests, I prayed alone. I was already doing that as a child and that has stayed with me. I light a candle now and then, but I don't attend Mass. My sister-in-law helped me a lot. I would call her, she would let me talk, she would listen to me.

Dialogue with God articulates the need to communicate, to ask questions and receive answers in order to attribute meaning to the traumatic event of illness. But it is a spiritual dialogue unmediated by the ecclesiastical institution, made up of a wholly personal rituality, which continues in the dialogue with the sister-in-law, whose help takes the form of the simple ability to listen attentively.

*2.2. The Acquisition of a Secular Spirituality*

The second form, which appears to be prevalent among the interviewees, by which latent spirituality is expressed is the acquisition of a new spiritual attitude, typically characterized by a secular and non-denominational orientation, and thus not linked to a specific denomination or religious affiliation. This form is present in the story of F.D. (a woman with breast cancer), in which the emergence of the spiritual dimension is connected to uncertainty for the future and fear of death:

> It took away some of my cheerfulness initially; I've always been a very joking, not very fatalistic person. What it has given me is that I used to say to myself "I don't have the money to eat pizza this month", but now I say to myself "I feel like eating a pizza and now I'm going to go". I see a T-shirt and I buy it. Today we are there and tomorrow we are gone. This aspect has opened my eyes. I have become perhaps more sensitive than I was, now I see a movie and I get emotional, before I used to tear up. Now I notice more of the things that happen around me, before life flowed ahead of me, I did things on a daily basis, now I dwell more. Now all the things I do I put more scruple into them, more thought into them, I'm more behind them. Before, I didn't dwell on what I was doing.

Death and dying, which in everyday experience remain abstract and distant concepts, with illness gain salience and become tangible concerns. The confrontation with end-of-life cracks the linearity of ordinary chronological time and heightens awareness of the value of time and the need to use it to the best advantage (Rasmussen and Elverdam 2007) as a "regained" time for self-realization. In F.D.'s story, illness thus opens the door to a different way of understanding spirituality as a personal and intimate need, to be experienced outside formal adherence to an organized religious tradition (Fuller 2001; Mercadante 2014; Parsons 2018):

> I am not a believer and so spirituality I don't know what spirituality means to me is a meaning that I can't understand I can't internalize spirituality is looking at nature and then there I feel at peace? I can only be at peace under certain conditions in the midst of greenery, nature. It is a condition that makes me feel good. I really envy those who believe I really envy them. And then I have to be honest, I also get quite annoyed sometimes by these attitudes, those who believe, these fatalisms. I believe a little bit in introspection, talking to oneself to get better if I want to solve something I have to talk to myself, I have to try to internalize things a little bit, I find the strength more inside me than outside, it's clear that in hard times it's hard to find it, the way I am help yourself that heaven help you, that speech is very valid. In fact if I think about the various gurus, hermits are always people who are alone with themselves, all these people who take refuge Saint Francis for example was with himself. The only time it stirred something inside me was when I went to see the cave of St. Francis in Umbria, a cave inside the rock. Well, I have to say that I found something spiritual there.

The spiritual quest is the focus of the narrative of F.S. (a man with lung cancer), whose experience of illness is pervaded by the need to connect to a deep dimension that gives meaning to his existence. The interviewee has practiced yoga for quite some time (although he stopped after the COVID-19 pandemic) while remaining dissatisfied with it:

> I have been going to a yoga school for ten years, which has not served me well, that is, yes and no. There is certainly a strong sensitivity towards the spiritual dimension. Yoga from that point of view is a great game because it makes you understand and sense that there's other stuff, but because I'm always kind of touristing all over the place, in the sense that I'm there and I'm not there, and I've done that with yoga, I've never applied myself to it more.

More than in a codified practice, such as yoga meditation, the interviewee perceives the spiritual dimension of illness and, more generally of existence, in ordinary situations of daily life:

The spiritual dimension definitely has an importance but I am better at perceiving it in the insubstantiality of what I see around. Something that for someone may have no meaning for me at that moment can take on an answer to those questions, which can be a scent, a light, a movement that makes you perceive an energy, a motivation, a beauty that is not intellectually understandable and narratable.

F.S.'s spiritual sensitivity was heightened by the onset of cancer:

[The illness] has opened up the pores a little bit more, although paradoxically I feel like it has completely closed them because I often get to thinking, yes, spirituality and all that is so not easily tangible, in a dimension like illness that is so grossly tangible, equals potential waste of time. So engaging in a certain kind of mindfulness, a certain kind of practice, may seem like a waste of time in the sense that from a practical point of view what's in it for me, what's the affective joviality in it for me? So the place it holds in my life is an odd place, coveted on one hand and when you least expect it felt on another.

It is through contact with nature, during his hikes in the mountains, that the interviewee is able to have experiences in a broadly mystical sense that allow him to grasp a deeper meaning in life and in illness itself:

Lately I've started going to the mountains again. You turn a path, you come across some stuff and there you feel all the power, let's call it energy, presence, and those moments are absolutely pregnant, very very intense moments of meaning, that maybe you don't reach by meditating for an hour on top of 2000 m. A little bit also when in life it catches you off guard, when you least expect the streetcar, and you get something that then are maybe basically the most sincere communications from the universe [...]. In those moments you are absolutely self-sufficient, you don't need anything, because it is so powerful what you have around you, you are so small but at the same time powerful, because you feel part of that power that you don't ask yourself anymore how big the mountain is, because it becomes one with yourself. That thing I consciously try to bring back into the everyday. That feeling there, that presence, brings me a quality of Fabio at that moment within the general existence of me and everything I see around.

Still, a secular type of spirituality, hinging on deep connection with nature, appeals M.R.T. (a woman with lung cancer), who has developed her own spiritual sensitivity as a result of illness:

I consider myself spiritual in the sense of being in nature, of appreciating nature in a different way. Since I have been ill, I seem to have heightened my senses much more. If I walk in the woods or walk among plants or even just look at the sky, I seem to appreciate it more because I see better, or maybe I see things I didn't notice before. Maybe I pay more attention to them, however, I appreciate them much more. I realize that I say very often, for example when I'm in the car: "but look at what a beautiful sky, but look at what beautiful plants, but look how green it is here, look how beautiful these leaves are, which before maybe I didn't". And that's something I like very much. I feel like I appreciate it much, much more.

In the story of S.A. (a woman with breast cancer) one can discern a markedly secular conception of spirituality that moves from an explicit awareness of the temporality of one's experience of illness:

For me it was really a matter of time, I would put an hourglass [...]. This is an awareness that I gained with time. Yes, time is long, however, it is not that then when it ends you go back and anyway this time in the meantime has passed, you have perhaps even aged, no? So it's not a parenthesis and you start from there, no, you don't start again, so it's not true that it stops, that it's a pause and you start again. From that moment it completely changes your life and it will never

be the same again [...]. So this time has passed, it has passed badly, because you go through a sickness both psychological and physical, and that time there is not going to give it back to you, but you can't complain because thank you that you had that time and you continue to have time, however that time there is no more, that person there is no more [...]. I was focused on my year-long journey. Actually one year is like ten all of a sudden, so you miss so many things, you miss so many pieces.

In this passage, the interviewee gives voice to an awareness that has matured over time and that has in time its own object of reflection. The years of illness profoundly affect the biographical path and mark it so radically that they imprint not only a clear turning point between the "before" and the "after", but a direction to the progression of life that appears to S.A. to be irreversible, "with no return" to the starting condition. Despite suffering, the interviewee acknowledges that she "cannot complain" about the fact that she was still able to have that time, following a "survivor model" that recurs frequently in cancer patients' histories (Hannum and Rubinstein 2016; Little et al. 2002). The time of illness is a heterotemporality that enters everyday time and permanently alters it, installing itself in biography as a caesura, as a "no more". This loss entails a temporal acceleration, because the duration of illness is asynchronous with respect to chronological duration, and "one year" is equivalent to "ten years passed at once". In light of this, the interviewee explains her own way of practicing spirituality, which is channeled toward alternative directions than institutional religion:

Seeing these comrades and friends of mine who did not deserve to die because they had fought so hard. I and others hated it when you would go to funerals and say "now finally rest in peace, amen". That's something I've always hated, because it goes on for 5, 10 years, but even just 2 years: do you do all that to rest in peace? No, that's not the purpose, otherwise one rests before suffering hell or one after a long battle, anyway he was sick for so many years, anyway all these years he spent them trying not to come to that end, so these things I always found very contradictory and annoying. Maybe I stuck more to the level of my spirituality as my own thing, maybe about the moon, about nature.

His is a pagan sensibility focused on the sacralization of nature that gradually emerges in the path of illness:

[The moon] has always fascinated me, it is something that relaxes me, almost as if it were a deity, but not a classical deity, I don't say that I am really pagan but I am more attached to many aspects, however I don't feel that I have the certainty of anything, so I am not one of those who has found or rediscovered God, taken refuge in God, no absolutely, because it seems to me an unfair thing to ask "save me and not save others", maybe there are those who deserve more, those who need more, those who suffer more [...]. I have always loved nature however not like this, this has changed. Maybe before I did not have the concept of it.

The inner self in this story becomes a discovery that occurs during the experience of cancer illness and transcends the conflict between the healthy self before and the sick self afterwards, pointing to a possible source of peace, well-being and meaning to overcome the trauma of cancer. A need for spirituality that is expressed as a search from open questions about existence, even before resting on certain answers, is discerned in the story of E.P. (a woman with breast cancer):

I never asked myself this question [about spirituality]. Unfortunately I think because of my lack of belief, to believe little, which I'm a little bit now reconsidering. I think however the people I have also crossed paths with a faith within, more than a Catholicism thing, with a serenity within, I think now I ask the question. The search though is always a bit of a search for those who don't believe. During the illness it was a secular search in toto, in the sense that I only saw reality, I didn't see the support of anything. Whereas now in hindsight and on reflection,

those who have this form of belief, of peace, have found in the search really an inner thing rather than a religion, I think absolutely they are lucky because the search is hard if you don't have hooks, and I am used to looking for hooks in myself more than outside.

The absence of these "hooks" outside of oneself is why the interviewee states that she has not prepared a time box, having experienced the disease as an entirely internal experience: "I have to rely so much on myself that I don't even cling to things".

N.F. (a man with lung cancer) is the only respondent who explicitly declares himself an atheist. In spite of this, even in his case, the disease provides a food for thought about life from which to discover new and unexpected aspects of himself:

I am an atheist, in the sense that I am a Christian but I don't go to church, so I don't profess, I didn't rely on religion. I would never have gone before to a doctor like P., because to me she was a shrink, And instead no, I have to say that by talking I discovered so many things [...]. I learned that the strength of our brain is very important, if we want we can. If we are strong from a mental point of view, we are also strong from a medical point of view. Very often our brain helps us to defeat an evil, not always, but very often it is willpower that is in charge. which I obviously did not have personally before [...]. It gave me the awareness of living a life, in the sense that today we are there and tomorrow we are not. I realized how important life is. I realized how important health is, money comes and goes [...]. I am quite fatalistic in the sense that I think each of us has a destiny already written. [Before the disease] I didn't think about it, before I was focused on making money, I didn't think about health because I had it.

### 3. Discussion—The Latent Dimension of Spirituality in Illness Narratives

The narratives analyzed in the preceding pages attest, in accordance with what the literature indicates about the spiritual care (Nissen et al. 2021), the relevance of the spiritual dimension in the biographies of the patients of the selected cancer associations. Despite the fact that the two groups of patients are affected by forms of cancer that have statistically different incidences in terms of life expectancy, their narratives document how the onset of cancer, whether breast or lung, marks a biographical disruption, a turning point between the phase before and the phase after diagnosis (Bury 1982). Following this moment, cancer patients' trajectories are marked by a second turning point, in which respondents gain a new existential perspective that allows them to derive reasons for personal learning and growth and to mature a different attitude toward their disease condition (Frank 1995; Williams 2000). To describe this new existential perspective, I use the term "spirituality" which, in line with what is suggested in the literature, constitutes a broader and more inclusive category than that of "religion". Spirituality, a multidimensional and polysemic concept (Fisher 2011) that pertains to cognitive, emotional, value-based and behavioral aspects (Paal et al. 2020), indicates the path of individual search for ultimate meaning, purpose, and transcendence and experiences of connection with the self, others, nature, and the sacred through beliefs, practices, and traditions (Best et al. 2023; Sulmasy 2002; Puchalski et al. 2014). Thus defined, spirituality is distinguished from religion which, instead, represents the organized, collective, and institutional form of the relationship with the sphere of the divine or the sacred (Hill et al. 2000). Although conceptually distinct, religion and spirituality establish different relationships, presenting themselves as opposing concepts (Heelas and Woodhead 2005) between those who declare themselves "spiritual but not religious" (Fuller 2001; Mercadante 2014; Parsons 2018) or mutually connected (Ammerman 2013), thus giving themselves a spirituality within religion or an extra-religious spirituality (Palmisano and Pannofino 2021).

In the stories collected, the interviewees express a typically spiritual attitude, centered on the authority of the subjective self, without affiliating themselves with the church or a formally organized group. For cancer patients, spirituality does not represent a mere preconstituted repertoire of symbols, values, and practices to draw on in order to cope

with suffering; rather, it turns out to be an unforeseen phenomenon that is formed and manifests itself during the course of treatment. Faced with an unforeseen and dramatic event such as a cancer diagnosis, in fact, previous certainties and existential references may undergo a profound reorganization and lead to an innovation in the individual's life path in which the sufferer manifests a latent spiritual capacity at first unexpressed that creatively adapts to the peculiar conditions posed by the pathology. In the narratives examined, latent spirituality manifests itself by taking two main forms: the rediscovery of the initial faith and the acquisition of a secular spirituality.

In the first case, it is a revitalization of Catholicism, the religion of belonging for all respondents; this return to the Catholic faith is mainly realized in the practice of individual prayer and devotion to a saint or, secondarily, in pilgrimage to sacred places and church attendance. Faith is thus experienced as a strengthening or rediscovery of feelings, values, or practices related to the faith of origin, which becomes a fundamental, though not exclusive, source of comfort during the most acute stages of illness. However, the illness can also weaken prior faith, leading to its personal reworking, which orients the sick person to positions that are more or less critical of institutionalized church religion, against which respondents opt for individual practice and inner dialogue with God, without the mediation of church hierarchies. In the second case, latent spirituality manifests itself as the acquisition of a new spiritual perspective, often secular and non-denominational in nature, which is nurtured by practices such as contact with nature, sports, introspection and yoga meditation, and beliefs such as that in the guardian angel or borrowed from Eastern philosophies and religions such as Hinduism and Buddhism or from pagan-type spiritualities.

As a phenomenon that emerges and takes shape along the course of treatment, latent spirituality must be analyzed in its temporal development. The stories of cancer patients illustrate how pathology translates from organic events into biographical experiences by configuring different ways of relating to time (Dolina et al. 2014; Robertson 2015). On the one hand, interviewees narrate their own identity change during the months or years of illness; on the other hand, they elaborate a discourse that has as its object the recognition of daily time and its existential value. In these narratives, the clinical time of the illness (Terenziani et al. 2008) is inscribed in the narrator's biography as another time or "heterotemporality" that pushes one to reorganize the rhythms and routines of daily life in relation to the commitments and needs imposed by the course of treatment. This time other produces estrangement effects whereby the patient tends to take on a new and different view of ordinary reality that may take on religious and spiritual coloring. Estrangement, which accompanies the transition from the familiar state of health to the unexpected and critical state of cancer illness, is a condition in which the subject must renegotiate the relationship with the body, the self, and significant others (Good 1994), as part of a representation of time marked by uncertainty and fear toward the future, the need to live the present as best as possible and the reinterpretation of the past as a period of normality (Broom et al. 2020). The alienating effects caused by the eruption of cancer into the biographical sphere solicit a spiritual vision with which the patient ascribes a value of sacredness to the time of daily life.

This spiritual interpretation of biographical time is expressed discursively in the form of a life review (Connolly and Timmins 2021; Haber 2006; Ng et al. 2022), and the recapitulation of past experience that is subjected to evaluative judgment in light of present living conditions and expectations for the future. In the stories examined, the life review involves an ambivalent evaluation of the past and the future. The past is interpreted as a period of normality to which one aspires to return but also, reread in retrospect, as a lost and inauthentic period, not fully experienced because it is regulated by work commitments and external rhythms. A lost or missing time is also the future, insofar as it is taken away by illness as a project phase of life. The balance between the past and the future, between what the disease has given and taken away, leads, in a seemingly paradoxical way, many patients interviewed to affirm that "since I've been ill, I live better", seeing in the present a regained and propitious moment to realize themselves and their desires. The reasons

given to explain this apparent paradox are the most diverse, but typically invoke a new sensitivity that favors, in the opinion of the patients themselves, to live with more depth, to enjoy more the "little things in life" and to value time, not in spite of the disease but because of it.

## 4. Materials and Methods

The material for the present article consists of a corpus of twenty qualitative interviews conducted using the creative time box technique and collected between October 2022 and May 2023 with Italian cancer patients. The study followed the logic of the "most different system design" (Cardano 2011; Seawright and Gerring 2008), a strategy of comparing two cases with different characteristics on a set of theoretically relevant variables in order to observe the recurrence of a common effect or behavior on another variable under investigation. In the context of this research, the cases are represented by two clinical settings with opposite characteristics on the level of disease experience, one consisting of breast cancer patients and the other of lung cancer patients, belonging to two associations operating in the area of the metropolitan city of Turin, in northern Italy: the Associazione Nazionale Donne Operate al Seno (ANDOS Onlus) at the Medical Oncology Department of Le Molinette Hospital and Women Against Lung Cancer in Europe (WALCE) at the Lung Oncology Department of San Luigi Gonzaga Hospital. The selected oncology patients (10 with breast cancer and 10 with lung cancer) correspond to statistically opposite clinical profiles in terms of life expectancy, with a more favorable prognosis for patients of the first type with breast neoplasm who, in Italy, survive in 88% of cases at 5 years after disease onset, compared with 19.5% on average among men and women for patients of the second type with lung neoplasm (AIOM 2022). The comparison between them was made from the preliminary hypothesis that the onset of severe disease generates questions or needs of a religious or spiritual nature from patients. In both groups, patients were recruited on a voluntary basis. In the case of lung cancer patients, the head of the WALCE association provided the list of patients, giving preference to subjects at the earliest stage of disease progression. In the case of patients with breast cancer, patients attended a meeting with the researcher at the ANDOS association, providing their willingness to participate in this study. In the case of the breast cancer patients, the selected sample is characterized by a diagnosis of the disease falling within the period 2011–2022, while in the case of the lung cancer patients, the diagnosis is on average more recent, falling within the period 2016–2022.

The interviews, conducted with patients both in-person and online, were structured around an outline consisting of three open-ended questions:

(a)  How has the disease changed your life?
(b)  How are you coping with this change?
(c)  What has the disease taken away from you and what has it given you?

These questions aim to delve deeper into the consequences of the oncological pathology on the life path and the spiritual meaning that the interviewees assign to the illness experience. To enable these questions to be answered, the interviews involved the use of the time box, a creative research method based on stimulus objects with the intention of encouraging participant engagement through a performative activity (Giorgi et al. 2021) that leverages the collaborative dimension of playful practice (Langley et al. 2022). The time box solicits autobiographical narrative of illness through the use of meaningful objects that the interviewee freely chooses to illustrate their experience. The object-mediated interview and the use of the time box allow, more than a narrative solely focused on verbal cues, the emotional and embedded expression of experience (Brown 2019; Fleetwood-Smith et al. 2021; Ravn 2021; Woodward 2015). The objects in the time box help interviewees construct, process and communicate the subjective meaning of their biographical story by articulating the illness narrative or pathography (Hawkins 1999; Nesby 2019), with respect to the past, present and future. The goal of the time box is thus to lead the patient to reflect on their own illness journey and the biographical transformation it produces over time. The temporal

dimension turns out to be the crucial aspect in understanding the spiritual implications of this experience and its processuality.

During the first contact, selected patients were asked to make the time box in preparation for the interview. A total of 14 of 20 interviewees agreed to the suggestion of constructing the time box to accompany the illness narrative during the interview. In some cases, however, interviewees had difficulty with this task, opting for a different solution, such as preparing cards with meaningful phrases or selecting only one object or, again, not utilizing any kind of object. The interviews, accompanied by photographs of the time boxes, were (video)recorded and later transcribed.

## 5. Conclusions

There is recognition in the nursing literature that spirituality is a resource of meaning to cope with suffering caused by the onset of chronic or terminal pathology (McSherry et al. 2021). This consideration underlies spiritual care, a caring approach focused on the relationship with the patient and their existential needs by physicians, nurses, or informal caregivers (Nissen et al. 2021), implemented especially in the field of palliative medicine (Best et al. 2023; Miller et al. 2023) in which it becomes more evident how illness affects not only the body, but the sick person in their biological, psychological, social, and spiritual wholeness (Batstone et al. 2020; Puchalski et al. 2019). Attention to spiritual needs is of essential importance in the care of patients with chronic or terminal illnesses such as cancer: failure to meet these needs can cause conditions of spiritual distress with repercussions on the patient's quality of life (Monod et al. 2010). Spiritual care, therefore, when integrated into clinical practice, can potentially be ameliorative in accompanying the sick person because it provides the patient with tools to attribute meaning to their illness experience, fostering active involvement along the therapeutic process of which they are a recipient (Taylor 2019; Yang et al. 2016).

The results of this article, while on the one hand confirming the relevance of the spiritual dimension in cancer pathology, on the other hand, in contrast to the prevailing approach in the medical humanities, allow us to emphasize that spirituality is not merely a resource of meaning to be drawn upon to overcome the suffering of illness, but is an emergent phenomenon along the therapeutic pathway and that it is transformed by reflecting the temporality of the biographical experience of illness: in fact, the collected narrative show how the condition of suffering can bring to light an unexpressed spirituality, consisting of the revitalization of previous traditional faith or the elaboration of an innovative lay spirituality. From the notion of latent spirituality as a phenomenon that emerges during the period of illness, it is possible to draw insights into some of the methodological limitations of the tools proposed in the nursing literature to investigate patients' spiritual needs and to design spiritual care interventions (Timmins and Caldeira 2017). In this literature, it is usually assumed that the spiritual dimension is a preconstituted repertoire of beliefs, meanings, or practices that the individual has and can draw on to implement coping strategies with respect to illness (Costa et al. 2019; Puchalski 2004). It is because of this assumption that scholars have developed screening instruments, especially questionnaires, to be administered at the beginning of the therapeutic course (Caldeira and Timmins 2017) in order to detect patients' religious and spiritual orientations through questions such as "do you consider yourself a religious person?", which appears as the first question in the FICA questionnaire (Puchalski 2014) or "what are the sources of your well-being?", in the opening of the HOPE questionnaire (Anandarajah and Hight 2001). The limitation of such an approach lies in losing the temporality that connotes the spiritual experience of illness (Toombs 1990), thus failing to monitor its change with a view to more effective spiritual care (McSherry et al. 2019). Attention to the subjective dimension of the time of illness can usefully complement the perspective of the biomedical model in which the time of illness experienced by the patient is transposed into the abstract and formal framework of hospital practice (Frankenberg 1988), a practice that involves coordination among health

care professionals, the use of technology, and the planning of interventions and procedures in order to optimize and standardize time of care (Pedersen and Obling 2020).

**Funding:** This research received no external funding.

**Institutional Review Board Statement:** The research undertaken did not require an institutional review according to the norms of the University of Turin.

**Informed Consent Statement:** Informed consent was obtained from all subjects involved in the study.

**Data Availability Statement:** The raw data supporting the conclusions of this article will be made available by the authors on request.

**Acknowledgments:** The author thanks Alessandro Comandone (Rete Oncologica Piemonte e Valle d'Aosta); Gruppo Italiano Tumori Rari (GITR); Fulvia Pedani (ANDOS—Le Molinette Hospital, Turin), Maria Vittoria Pacchiana, and Silvia Novello (WALCE—San Luigi Gonzaga Hospital, Orbassano) for their collaboration in the research and for providing contacts with the interviewees.

**Conflicts of Interest:** The author declares no conflicts of interest. The funders had no role in the design of the study; in the collection, analyses, and interpretation of data; in the writing of the manuscript; and in the decision to publish the results.

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
