# Peer review of "“Since I’ve Been Ill, I Live Better”: The Emergence of Latent Spirituality in the Biographical Pathways of Illness"

_religions, doi:10.3390/rel15010090_

Round 1

Reviewer 1 Report

Comments and Suggestions for Authors

Thank you for the opportunity to review an article that I read with interest and which addresses the critical area of latent spirituality in the life of a sick person.

The manuscript makes many important observations and, as such, points to spirituality as a critical resource for (not only) the ill person.

However, while reading the article, a few thoughts or uncertainties came to mind, and with all due respect to their work, I would ask the authors.

In the Introduction, it would be helpful to organize the paragraphs better so that certain information is not repeated in different places.

What occurs in lines 81-96 is more in the Introduction of the article, where it talks about the need for spirituality.

If the entire Introduction is titled "The latent dimensions of spirituality," I would expect to learn what latent spirituality is proper in the Introduction; this way, the information disappears amongst much other information. Is the title even necessary? The Introduction also talks about other matters.

It would seem appropriate to reword the introduction's last paragraph to include the study's aim and reasons. The results here (lines 77-81, 97-102) would be more appropriately relocated to the Results section.

I wonder why line 61 says in sociology? What about the other areas? Health psychology, theology, etc.

In section 2, Materials and Methods, I don't understand the expression in line 108, "the most different system design."? Do the authors mean a "completely" different design? The phrase the most different is a bit confusing and asks for clarification.

Were there any criteria for inclusion of patients in the study other than the cancer diagnosis mentioned? For example, diversity in terms of gender, age, length of illness, education, or socioeconomic background?

I would expect information on how patients were recruited to the study and whether they consented and had any benefits from participating.

What does the statement on line 150, "most interviewees," mean? How many were there, at least in percentage terms?

In section 3 Results, I'm thinking again about our section. What are spiritual stories? Aren't they stories about spirituality?

You state that these are some illness narratives (l159) and selected illness narratives (l164). Why were these chosen?

I am also thinking about section 3.1, The rediscovery of prior religiosity. I think the authors are more concerned with latent spirituality, and religiosity itself is a somewhat different concept (see Zinnbauer, Pargament, Ammeman, etc.). So why do they see the rediscovery of prior religiosity as the discovery of latent spirituality?

Moreover, for some patients (T.S., M.V.), it is instead a questioning of religiosity or a departure from it. In contrast, for others (e.g., T.S., P.B.), the rediscovery of prior religiosity is described more as a leaning towards the help of saints and relying on the power of medallions and amulets. Is this really spirituality or rather superstition and seeking any use without a personal relationship with God?

Some summary at the end of section 3.1 of what this rediscovery of prior religiosity really means for patients about their spirituality would be helpful.

Otherwise, their conclusions are similar to what is in section 3.2 as a discovery of secular spirituality. E.g., for a P.F. woman with breast cancer - why is she in part 3.2 and not in 3.1 when her story is similar to some of those in part 3.1?

Similarly, it is appropriate to specify this in the discussion, lines 602 and nn. I am not sure there has been a revitalization of Catholicism and a return to the Catholic faith in the patients in section 3.1. Or is there just a move towards certain practices (devotion to saints, pilgrimages), which are not the essence of the Catholic faith but only secondary manifestations?

In the conclusion itself, apart from the limits of the thesis, I would have welcomed some recommendations for practice or a final summary so that the conclusion is really a conclusion of what the research has brought.

I thank the authors for their significant research and believe that my minor comments, once accepted, could contribute to the clarity of the interpretation of their interesting findings.

Comments on the Quality of English Language

Sometimes, it is a bit difficult to understand your formulations, especially in the parts where they are transcripts of interviews. I know that it is a recording of direct speech, but sometimes it is harder to understand,

Author Response

Response to Reviewer 1 Comments

Thank you very much for taking the time to review this manuscript and for the precise and constructive comments on the text. Please find the detailed responses below and the corresponding revisions/corrections in the re-submitted file. 

Questions for General Evaluation

Reviewer’s Evaluation

Does the introduction provide sufficient background and include all relevant references?

Can be improved

Are all the cited references relevant to the research?

Yes

Is the research design appropriate?

Must be improved

Are the methods adequately described?

Must be improved

Are the results clearly presented?

Can be improved

Are the conclusions supported by the results?

Can be improved

Point-by-point response to Comments and Suggestions for Authors

Comments 1: Thank you for the opportunity to review an article that I read with interest and which addresses the critical area of latent spirituality in the life of a sick person.

The manuscript makes many important observations and, as such, points to spirituality as a critical resource for (not only) the ill person.

However, while reading the article, a few thoughts or uncertainties came to mind, and with all due respect to their work, I would ask the authors.

In the Introduction, it would be helpful to organize the paragraphs better so that certain information is not repeated in different places.

What occurs in lines 81-96 is more in the Introduction of the article, where it talks about the need for spirituality.

If the entire Introduction is titled "The latent dimensions of spirituality," I would expect to learn what latent spirituality is proper in the Introduction; this way, the information disappears amongst much other information. Is the title even necessary? The Introduction also talks about other matters.

Response 1: Thank you for pointing this out. I agree with this comment. Therefore, I removed from the Introduction the considerations relating to the theme of spiritual care (which were found in the lines 81-96). This part is now found in the Conclusions (see lines 624-635).

I also removed the title from the Introduction.

Comments 2: It would seem appropriate to reword the introduction's last paragraph to include the study's aim and reasons. The results here (lines 77-81, 97-102) would be more appropriately relocated to the Results section.

Response 2: Agree. I revised the final paragraph of the introduction, indicating the aim of the study and adding a synopsis of the article (see lines 76-86). 

The results in the lines 97-102 have now been moved to the Results section (see lines 635-639).

Comments 3: I wonder why line 61 says in sociology? What about the other areas? Health psychology, theology, etc.

Response 3: I also added the reference to medical humanities (see line 47).

Comments 4: In section 2, Materials and Methods, I don't understand the expression in line 108, "the most different system design."? Do the authors mean a "completely" different design? The phrase the most different is a bit confusing and asks for clarification.

Response 4: “most different system design” is the full name of the method. It is, as indicated in the section, a method of comparison between empirical cases that have opposite characteristics. in my case, these are two groups of patients suffering from forms of cancer who have statistically opposite clinical outcomes (relatively low life expectancy for patients with lung cancer and relatively high life expectancy for patients with breast cancer). In this sense, it differs from the complementary method, called "most similar system design". with which a comparison is made between empirical cases with identical characteristics (for example, two groups of patients suffering from the same type of cancer).

Comments 5: Were there any criteria for inclusion of patients in the study other than the cancer diagnosis mentioned? For example, diversity in terms of gender, age, length of illness, education, or socioeconomic background?

I would expect information on how patients were recruited to the study and whether they consented and had any benefits from participating.

Response 5: I added this information in section 2 (see lines 110-117).

Comments 6: What does the statement on line 150, "most interviewees," mean? How many were there, at least in percentage terms?

Response 6: I indicated the exact number (14 out of 20) in section 2 (see line 144).

Comments 7: In section 3 Results, I'm thinking again about our section. What are spiritual stories? Aren't they stories about spirituality?

You state that these are some illness narratives (l159) and selected illness narratives (l164). Why were these chosen?

Response 7: I standardized, also in this case using the expression “illness narratives”.

The selection criterion for these narratives is indicated at the beginning of paragraph 3 (see lines 153-159).

Comments 8: I am also thinking about section 3.1, The rediscovery of prior religiosity. I think the authors are more concerned with latent spirituality, and religiosity itself is a somewhat different concept (see Zinnbauer, Pargament, Ammeman, etc.). So why do they see the rediscovery of prior religiosity as the discovery of latent spirituality?

Moreover, for some patients (T.S., M.V.), it is instead a questioning of religiosity or a departure from it. In contrast, for others (e.g., T.S., P.B.), the rediscovery of prior religiosity is described more as a leaning towards the help of saints and relying on the power of medallions and amulets. Is this really spirituality or rather superstition and seeking any use without a personal relationship with God?

Some summary at the end of section 3.1 of what this rediscovery of prior religiosity really means for patients about their spirituality would be helpful.

Otherwise, their conclusions are similar to what is in section 3.2 as a discovery of secular spirituality. E.g., for a P.F. woman with breast cancer - why is she in part 3.2 and not in 3.1 when her story is similar to some of those in part 3.1?

Similarly, it is appropriate to specify this in the discussion, lines 602 and nn. I am not sure there has been a revitalization of Catholicism and a return to the Catholic faith in the patients in section 3.1. Or is there just a move towards certain practices (devotion to saints, pilgrimages), which are not the essence of the Catholic faith but only secondary manifestations?

Response 8: To respond more adequately to these observations, I have added to the Discussion a part in which I address the definition of the concept of spirituality and its relationship with that of religion, on the basis of the prevailing literature in the sociology of religion and in nursing sciences on spiritual care (see lines 541-556). 

On the basis of these definitions, it is possible to consider all the illness narratives analyzed as examples of spirituality, both those that rediscover the original Catholic faith and those that acquire a secular spiritual vision. In the first case, in particular, the return to Catholicism always occurs through a more or less critical personal re-elaboration which places the emphasis on the subjective dimension of the experience of faith rather than on formal belonging to the church institution and on adherence to dogmas and beliefs (a characteristic, this, typically attributable to the concept of spirituality). Moreover, precisely the return to Catholicism is, from my point of view, the main indication of the existence of a latent spirituality, or rather of a propensity towards the religious that pre-exists the event of the illness but which manifests itself only as a consequence of this.

Two of the interviews cited as problematic - the interview with M.V. in section 3.1 and with P.F. in section 3.2 - were eliminated, also in response to the suggestion of other reviewers who recommended shortening section 3.

Comments on the Quality of English Language

Sometimes, it is a bit difficult to understand your formulations, especially in the parts where they are transcripts of interviews. I know that it is a recording of direct speech, but sometimes it is harder to understand.

Response: I revised the language in the interview excerpts. I intervened at the critical points where I had carried out a literal transcription of the speech, favoring a more grammatically correct translation

ERRATA CORRIGE
The correct title of section 3 is: Results. The spiritual illness narratives of cancer patients
(and not, as erroneously written in the re-submitted file: Results. The spiritual illness of cancer patients).

Reviewer 2 Report

Comments and Suggestions for Authors

This is an  outstanding piece of research with a great title to introduce it: Since I've been Ill,  I live better: The emergence of latent spirituality in biographical pathways of illness.

1. the descriptions in the abstract are amazing, such as " latent spiritualty and "tacit propensity; condition of suffering can bring to light an unexpressed spirituality;  pathology constitutes a biographical fracture, accompanied by questions and needs of a religious and spiritual nature.

The introduction centers in on some key points, such as the opening: Spirituality can be a crucial resource to draw on to make sense of critical situations  that mark a turning point in individual and collective biographies 

Suggestion 1.  put   #30.....statement " recourse to spirituality can open up perspectives  of meaning by which disaster survivors overcome the suffering of the present moment  and re-establish a state of well-being by leveraging feelings of hope and group cohesion"   AND 60  "  The latent dimension of spirituality, in this sense, has so far received little attention in sociology........  ( these points should be right at the beginning of the INTRODUCTION  to EMPHASHIZE  and SUPPORT THE PURPOSE OF YOUR RESEARCH. 

63 great quote : no atheists in fox holes"

78: great statement: suffering can bring to light an unexpressed spirituality.

The materials and methods are very clear!

     There are good questions: how has the disease changed your life?1 (b) how are you coping with this change?  (c) what has the disease taken away from you and what has it given you. 

         These are good open questions. 

Also,  the time box was a good instrument to apply to the methods .

The spiritual stories selected for the results are amazing and support the research. 

1. [The illness] has given me the opportunity to live more quietly, with my family, live more 220 outdoors, it has given me a better life in that sense, I have found benefit, freer, less stressful. 

     The discussion is communicated very clearly.

594-5 was a strong statement:  it turns out to be an unforeseen phenomenon that is formed and manifests itself during  the course of treatment. 

* This again is supportive of the findings, related to an understanding of spirituality "

*     SUGGESTION:   Some of the long paragraphs should be shortened, to make the main points stand out better.

 IN conclusion: This a a very high quality piece of scholarly research that deserves to be published!  Keep up the good work. 

Author Response

Response to Reviewer 2 Comments

Thank you very much for taking the time to review this manuscript, and for your useful comments and suggestions. Please find the detailed responses below and the corresponding revisions/corrections in the re-submitted files. 

Questions for General Evaluation

Reviewer’s Evaluation

Does the introduction provide sufficient background and include all relevant references?

Yes

Are all the cited references relevant to the research?

Yes

Is the research design appropriate?

Yes

Are the methods adequately described?

Yes

Are the results clearly presented?

Yes

Are the conclusions supported by the results?

Yes

Point-by-point response to Comments and Suggestions for Authors

Comments 1: This is an  outstanding piece of research with a great title to introduce it: Since I've been Ill,  I live better: The emergence of latent spirituality in biographical pathways of illness.

1. the descriptions in the abstract are amazing, such as " latent spiritualty and "tacit propensity; condition of suffering can bring to light an unexpressed spirituality;  pathology constitutes a biographical fracture, accompanied by questions and needs of a religious and spiritual nature. 

The introduction centers in on some key points, such as the opening: Spirituality can be a crucial resource to draw on to make sense of critical situations  that mark a turning point in individual and collective biographies 

Suggestion 1.  put   #30.....statement " recourse to spirituality can open up perspectives  of meaning by which disaster survivors overcome the suffering of the present moment  and re-establish a state of well-being by leveraging feelings of hope and group cohesion"   AND 60  "  The latent dimension of spirituality, in this sense, has so far received little attention in sociology........  ( these points should be right at the beginning of the INTRODUCTION  to EMPHASHIZE  and SUPPORT THE PURPOSE OF YOUR RESEARCH.

Response 1: Thank you for pointing this out. I modified it as follows:

In the introduction I anticipated the part relating to the concept of latent spirituality, changing the order of the paragraphs. After the sentence “recourse to spirituality can open up… et seq.”, I inserted the sentence “These studies agree that spirituality is a resource…” (see lines 38-47), postponing the part relating to COVID (starting from line 48). 

Comments 2: 

63 great quote : no atheists in fox holes"

78: great statement: suffering can bring to light an unexpressed spirituality.

The materials and methods are very clear!

There are good questions: how has the disease changed your life?1 (b) how are you coping with this change?  (c) what has the disease taken away from you and what has it given you. 

These are good open questions. 

Also,  the time box was a good instrument to apply to the methods.

The spiritual stories selected for the results are amazing and support the research. 

1. [The illness] has given me the opportunity to live more quietly, with my family, live more outdoors, it has given me a better life in that sense, I have found benefits, freer, less stressful.

The discussion is communicated very clearly.

594-5 was a strong statement:  it turns out to be an unforeseen phenomenon that is formed and manifests itself during  the course of treatment. 

* This again is supportive of the findings, related to an understanding of spirituality"

Response 2: I thank the Reviewer for appreciation of these parts of the article.

Comments 3: SUGGESTION: Some of the long paragraphs should be shortened, to make the main points stand out better.

Response 3: I agree with this suggestion. Therefore, I eliminated two interview excerpts, one from paragraph 3.1 and the other from paragraph 3.2, to shorten the overall length of section 3

ERRATA CORRIGE
The correct title of section 3 is: Results. The spiritual illness narratives of cancer patients
(and not, as erroneously written in the re-submitted file: Results. The spiritual illness of cancer patients).

Reviewer 3 Report

Comments and Suggestions for Authors

The article deals with a timely topic and this needs to get the attention. I have a few comments to make that would improve the article as presented:

- Under 2. Materials and methods, line 134ff > make sure to make the objects explicit as they relate heavily to the religious meaning construction in the paper.

- There is no explicit definition of religiosity/spirituality and this is problematic; this needs to be improved by stating clearly what types of criteria are used; also a stronger connection with existing literature needs to be added.

- Section 3 is lengthy with the two types of spirituality; I would recommend 2-3 representative citations and then elaborate on those.

Author Response

Response to Reviewer 3 Comments

Thank you very much for taking the time to review this manuscript. Please find the detailed responses below and the corresponding revisions/corrections in the re-submitted file.

Questions for General Evaluation

Reviewer’s Evaluation

Does the introduction provide sufficient background and include all relevant references?

Can be improved

Are all the cited references relevant to the research?

Yes

Is the research design appropriate?

Can be improved

Are the methods adequately described?

Can be improved

Are the results clearly presented?

Can be improved

Are the conclusions supported by the results?

Can be improved

Point-by-point response to Comments and Suggestions for Authors

Comments 1: The article deals with a timely topic and this needs to get the attention. I have a few comments to make that would improve the article as presented:

- Under 2. Materials and methods, line 134ff > make sure to make the objects explicit as they relate heavily to the religious meaning construction in the paper.

Response 1: Thank you for pointing this out. Agree. Therefore, I explained the object of the study, adding a sentence in which I specify the purpose of the questions that make up the interview with cancer patients (see lines 125-127).

Comments 2: There is no explicit definition of religiosity/spirituality and this is problematic; this needs to be improved by stating clearly what types of criteria are used; also a stronger connection with existing literature needs to be added.

Response 2: Agree. I have, accordingly, strengthened this point, focusing on the definitions of spirituality and religion, and on the relationships that link the two concepts. For each of these aspects, I have integrated further bibliographical references, taken from the literature in the sociology of religion and in nursing sciences on the theme of spirituality in care (see lines 541-559). 

Comments 3: Section 3 is lengthy with the two types of spirituality; I would recommend 2-3 representative citations and then elaborate on those

Response 3: I agree with this suggestion. Therefore, I eliminated two interview excerpts, one from paragraph 3.1 and the other from paragraph 3.2, to shorten the overall length of section 3

ERRATA CORRIGE
The correct title of section 3 is: Results. The spiritual illness narratives of cancer patients
(and not, as erroneously written in the re-submitted file: Results. The spiritual illness of cancer patients).

Reviewer 4 Report

Comments and Suggestions for Authors

very interesting and well presented paper, particularly useful for caregivers

Author Response

Response to Reviewer 4 Comments

Thank you very much for taking the time to review this manuscript. Please find the detailed responses below and the corresponding revisions/corrections in the re-submitted file.

Questions for General Evaluation

Reviewer’s Evaluation

Does the introduction provide sufficient background and include all relevant references?

Yes

Are all the cited references relevant to the research?

Yes

Is the research design appropriate?

Yes

Are the methods adequately described?

Yes

Are the results clearly presented?

Yes

Are the conclusions supported by the results?

Yes

Point-by-point response to Comments and Suggestions for Authors

Comments 1: Very interesting and well presented paper, particularly useful for caregivers

Response 1: I thank the Reviewer for appreciating my article and for noting its possible application in care practice by caregivers

ERRATA CORRIGE
The correct title of section 3 is: Results. The spiritual illness narratives of cancer patients
(and not, as erroneously written in the re-submitted file: Results. The spiritual illness of cancer patients).

Round 2

Reviewer 1 Report

Comments and Suggestions for Authors

Thanks for the careful review and answering my questions. Congratulations on a nice article.